# LncRNA-Profile-Based Screening of Extracellular Vesicles Released from Brain Endothelial Cells after Oxygen–Glucose Deprivation

**DOI:** 10.3390/brainsci12081027

**Published:** 2022-08-03

**Authors:** Xiang He, Hecun Zou, Qiang Lyu, Yujing Tang, Wenkui Xiong, Fei-Fei Shang

**Affiliations:** 1Department of Anesthesiology, Guizhou Provincial People′s Hospital, Guiyang 550002, China; hexgmc@163.com; 2Institute of Life Science, Chongqing Medical University, Chongqing 400016, China; zouhecun@cqmu.edu.cn (H.Z.); tang_yujing2000@126.com (Y.T.); xiongwenkui@163.com (W.X.); 3Department of Anesthesiology, Sichuan Provincial People′s Hospital, University of Electronic Science and Technology of China, Chengdu 610072, China; 15608023302@163.com

**Keywords:** lncRNAs, human brain microvascular endothelial cells, cerebral ischemia, extracellular vesicles

## Abstract

Brain microvascular endothelial cells (BMECs) linked by tight junctions play important roles in cerebral ischemia. Intercellular signaling via extracellular vesicles (EVs) is an underappreciated mode of cell–cell crosstalk. This study aims to explore the potential function of long noncoding RNAs (lncRNAs) in BMECs’ secreted EVs. We subjected primary human and rat BMECs to oxygen and glucose deprivation (OGD). EVs were enriched for RNA sequencing. A comparison of the sequencing results revealed 146 upregulated lncRNAs and 331 downregulated lncRNAs in human cells and 1215 upregulated lncRNAs and 1200 downregulated lncRNAs in rat cells. Next, we analyzed the genes that were coexpressed with the differentially expressed (DE) lncRNAs on chromosomes and performed Gene Ontology (GO) and signaling pathway enrichment analyses. The results showed that the lncRNAs may play roles in apoptosis, the TNF signaling pathway, and leukocyte transendothelial migration. Next, three conserved lncRNAs between humans and rats were analyzed and confirmed using PCR. The binding proteins of these three lncRNAs in human astrocytes were identified via RNA pulldown and mass spectrometry. These proteins could regulate mRNA stability and translation. Additionally, the lentivirus was used to upregulate them in human microglial HMC3 cells. The results showed NR_002323.2 induced microglial M1 activation. Therefore, these results suggest that BMECs’ EVs carry the lncRNAs, which may regulate gliocyte function after cerebral ischemia.

## 1. Introduction

Ischemic stroke is caused by the obstruction of a cerebral artery, leading to a sudden decrease in the blood supply and, thus, hypoxia, nutrient deficiency, and metabolic waste accumulation [1]. In mammals, vascular endothelial cells form tight junctions of blood–brain barrier (BBB), which is selectively permeable [2]. Endothelial cells express specific transporters to transport nutrients such as glucose while secreting neurotrophic factors and proteases to nourish nerve cells or degrade harmful molecules [3]. Under adverse stimuli, the overreaction of vascular endothelial cells can damage the integrity of the BBB and aggravate injury. For example, the Px1 channel of vascular endothelial cells is activated and, therefore, releases ATP, inflammatory factors, and other signaling molecules to send “find-ME” signals to inflammatory cells after cerebral ischemic injury, thus attracting lymphocytes and aggravating inflammatory responses [4,5]. Therefore, endothelial cells play important roles in responding to external stimuli and maintaining the homeostasis of the brain, in which the secretion function is an important means whereby endothelial cells respond to stimuli.

The intercellular connections observed within organisms are very tight. Signals are transmitted between cells by means of signaling proteins, hormones, and vesicles. Extracellular vesicles (EVs) produced by cells have been found to play important roles in the transmission of information between cells. EVs released by cells in normal and stressed states are balloon-like structures surrounded by lipid bilayers [6]. EVs contain cytokines, growth factors, and genetic material such as coding and noncoding RNAs, as well as other cytoplasmic components and molecules [7]. Data have shown that endothelial cells secrete significantly different EVs under adverse stimuli such as hypoxia, inflammation, and high glucose levels. miR-214 in endothelial-cell-derived exosomes has been found to promote angiogenesis in vitro and in vivo by inhibiting the expression of ataxia-telangiectasia mutated gene [8]. Studies of cerebral ischemic injury have found that EVs secreted by cerebral microvascular endothelial cells under TNFα stimulation contain a large number of NFκB pathway proteins and mitochondrial-associated proteins [9]. In conclusion, cerebral microvascular endothelial cells can attract inflammatory cells by secreting EVs upon cerebral ischemia and aggravate injury. It is important to explore the EVs secreted by cerebral microvascular endothelial cells under ischemic conditions.

Noncoding RNAs are an important class of regulatory molecules that have been repeatedly confirmed to exist in EVs [10,11,12]. Among the different types of noncoding RNAs, (long noncoding RNAs) lncRNAs are transcribed by RNA polymerase, have a length of ≥200 nucleotides, and do not include a coding sequence [13]. Studies have shown that lncRNAs are involved in the regulation of a variety of diseases, including tumors, cardiovascular system diseases, nervous system diseases, and immune system diseases. They can also regulate gene expression at multiple levels, including regulation via epigenetic mechanisms, such as DNA methylation and genomic imprinting; transcriptional mechanisms, such as binding with transcription factors to control gene transcription activity; and posttranscriptional mechanisms, such as binding with miRNA as an endogenous “miRNA sponge”, thereby affecting the regulation of target genes by miRNAs [14]. In a study of cerebral ischemic injury, Michalik et al. found that endothelial cells were damaged after cerebral ischemia and that specific lncRNAs, such as linc00493, Meg3, and MALAT1, were increased in vascular endothelial cells. Furthermore, after knocking out MALAT1 in mice, it was found that the cell cycle of vascular endothelial cells was inhibited, the number of cells decreased, and cells migrated, sprouted, and blocked new blood vessels. It can be inferred that MALAT1 is involved in the regulation of angiogenesis [15]. Therefore, lncRNAs in EVs released by vascular endothelial cells may be the key to revealing the mechanism of secondary injury induced by cerebral ischemia.

The concept of a neurovascular unit emerged from the fact that the blood–brain barrier is composed of a variety of cell types and emphasizes the dynamic interaction between endothelial cells, glial cells, and neurons to build neurovascular units in the microenvironment [16]. This increases the importance of the EVs responsible for intercellular signaling. In this study, we subjected human and rat primary brain microvascular endothelial cells to oxygen and glucose deprivation (OGD) as described above, followed by the collection of EVs in a cell culture medium, to explore the potential function of lncRNAs (Figure 1).

## 2. Methods

### 2.1. Cell Culture

The OGD experiment was performed on human primary cerebral microvascular endothelial cells (BMECs) according to our previously reported experimental method, and passage 1 cells were purchased from the Angio-Proteomie Company(product number cAP-0002, Boston, MA, USA). Endothelial cell culture medium (ECM, ScienCell, Carlsbad, CA, USA) was used to culture and passage BMECs to the fifth passage. Oxygen–glucose deprivation was performed using a glucose-free M199 medium (Thermo Fisher, Waltham, MA, USA) with a final formulation of 20% fetal bovine serum (FBS) (Thermo Fisher, Waltham, MA, USA), 20 µg/mL endothelial cell growth supplement (ECGS) (Merck Millipore, Burlington, MA, USA), 0.1 mg/mL sodium heparin (Thermo Fisher, Waltham, MA, USA), 2 mmol/L glutamine (Thermo Fisher, Waltham, MA, USA), and 1% penicillin–streptomycin (Thermo Fisher, Waltham, MA, USA). The FBS was centrifuged at 100,000× *g* for 18 h to remove EVs. The OGD model was set up in a three-gas incubator with 2% oxygen and glucose-free M199 for 24 h, followed by the recovery of oxygen and glucose for another 24 h prior to the next experiment. Primary astrocytes were purchased from ScienCell Company (product number 1800, Carlsbad, CA, USA) and cultured using DMEM with 20% FBS.

Rat primary brain microvascular endothelial cells were also obtained according to our previous experimental method. Eight adult (6-week-old) male Sprague Dawley rats weighing 160–200 g were used. The rats were housed in a specific pathogen-free (SPF) animal room at the Animal Breeding Center of Chongqing Medical University. All animal methods were approved by the Chongqing Medical University Committee on Animal Research, China (approval No. CQMU20200133) on 11 November 2020. Rats were anesthetized with isoflurane (2% oxygen) (RWD Life Science, Shenzhen, China) and then euthanized, and their brains were collected. After the brains were homogenized, the homogenates were centrifuged at 720× *g* for 5 min at 4 °C, and the pellets were retained. They were then resuspended in phosphate-buffered saline (PBS), and these samples were transferred to 15 mL of 18% dextran and stratified via centrifugation at 4500× *g* for 20 min at 4 °C. The cells in the middle layer were collected and resuspended in 10 mL of PBS containing 0.1% bovine serum albumin, followed by the addition of 100 µL of collagenase (100 mg/mL), 40 µL of DNase I (10 mg/mL) and 100 µL of N-alpha-tosyl-L-lysine chloromethyl ketone hydrochloride (TLCK) (14.7 µg/mL), gentle shaking and mixing, and digestion at 37 °C for 1 h. The cell pellet was then collected via centrifugation at 1000× *g* for 5 min. The cells were incubated with 2 mL of a biotin-labeled CD31 antibody (DSB-X Biotin Protein Labeling Kit, Thermo Fisher, Waltham, MA, USA) in 0.1% bovine serum albumin in PBS for 10 min at 4 °C. Then, streptavidin-coupled magnetic beads (Dynabeads FlowComp Flexi Kit, Waltham, MA, USA, Thermo Fisher) were used to enrich CD31-positive BMECs. BMECs were identified by the immunofluorescence detection of the endothelial cell markers factor VIII and CD31 (Thermo Fisher, Waltham, MA, USA). Finally, the cells were cultured with ECM as described above, and fifth-passage cells were subjected to OGD experiments.

### 2.2. EV Isolation

Cell supernatants were collected (80 mL per group) and centrifuged at 300× *g* for 10 min at 4 °C. The resultant supernatants were collected and centrifuged again at 3000× *g* for 20 min. The obtained supernatants were centrifuged at 10,000× *g* for 30 min, and the resultant supernatants were collected and centrifuged at 100,000× *g* for 70 min at 4 °C. The obtained pellet was resuspended in 1 mL of PBS. The samples were centrifuged again at 100,000× *g* for 70 min, and the pellet was finally resuspended in 100 μL PBS to obtain EVs.

### 2.3. RNA Sequencing

TRIzol reagent (Thermo Fisher, Waltham, MA, USA) was used to extract EV RNAs. After washing with precooled 75% ethanol, RNA samples were centrifuged at 12,000 rpm at 4 °C. The supernatant was discarded, and 50 µL of RNase-free water was added to dissolve the RNA. The purity of the total RNA was assessed on a Nanodrop system (Thermo Fisher, Waltham, MA, USA). The OD260/280 ratio was used to evaluate RNA degradation and protein contamination in the extracted RNA. The OD260/230 was used to evaluate contamination by organic matter or salt ions. Qualified RNA samples were depleted of ribosomal RNA (rRNA) from total RNA using an Epicentre Ribo-Zero Kit (Epicentre, Madison, WI, USA), followed by the purification of the RNA (RNeasy MinElute Cleanup Kit, QIAGEN, Germantown, MD, USA). The purified RNA was randomly fragmented into short fragments using a fragmentation reagent (RNA Library Prep Master Mix, NEB, Ipswich, MA, USA). Next, the RNA was fragmented as a template, random primers were used to synthesize single-stranded cDNA, and dNTPs, DNA Polymerase I, and RNaseH were successively added to synthesize double-stranded cDNA. Afterward, the double-stranded product was purified using AMPureXP (NEB, Ipswich, USA), and the sticky ends of the DNA were transformed into blunt ends by using DNA Polymerase I, Large (Klenow) Fragment, and T4 DNA polymerase (NEB, Ipswich, USA). A nucleobase A and a linker were added to the 3′ end, and the second strand of the cDNA containing U was then degraded by Thermolabile USER Enzyme (NEB, Ipswich, USA). Next, PCR amplification was performed to generate a sequencing library. Quantification was first performed using a Qubit 2.0 Fluorometer (Thermo Fisher, USA). The constructed library was then diluted to a concentration of 1 ng/µL. Thereafter, the insert size of the library was detected using an Agilent 2100 Bioanalyzer (Agilent, Santa Clara, CA, USA), and qPCR was applied for precise quantification. After the library passed the quality inspection, the Illumina HiSeq 4000 platform was used for sequencing, and the sequence read length was set as follows: double-ended 2 × 125 bp.

### 2.4. Sequencing Data Analysis

After obtaining raw sequencing data (raw data), the data were filtered, and low-quality reads were processed to obtain high-quality data (clean data). HISAT2 software was used to compare the clean data with the reference genome and to comprehensively evaluate the coverage area and coverage depth of the sequencing data. The reads aligned to the chromosomes were annotated as exonic (exons), intronic (introns), or intergenic (intergenic regions). Next, we evaluated whether the newly detected genes showed decreasing abundance with the increase in the volume of sequencing data to determine whether the data were sufficient. The expression level of each lncRNA was then calculated as the expected number of reads per kilobase of transcript sequence per million base pairs sequenced (RPKM). DE lncRNA levels were calculated using DESeq2 based on the negative binomial generalized linear model, and lncRNAs with a fold change greater than 1.5 were selected as DE lncRNAs and displayed in the heatmap, where the *p*-value refers to the probability of an extreme case under the hypothesis that all genes show no difference between the two experimental conditions. The Q value was obtained by correcting the *p*-value using the DESeq2 algorithm, where a lower Q value indicated a more significant difference in gene expression.

lncRNAs may have regulatory effects on adjacent protein-coding genes. We set the colocalization threshold to 100 kb upstream and downstream of the lncRNA. Subsequently, the main functions of the lncRNAs were predicted through the functional enrichment analysis of mRNA genes showing colocalization with lncRNAs. The biological pathway analysis was based on the Kyoto Encyclopedia of Genes and Genomes (KEGG) biological pathway database (http://www.genome.jp/, accessed on 10 March 2020); in this analysis, the biological pathways of gene sets were analyzed from the perspective of complex regulatory networks. Gene Ontology (GO) analysis can be performed to annotate the function of each gene and calculate the most significant function in a specific series of genes based on the hypergeometric distribution. The analysis results are shown as bubble diagrams. The number of genes in each category divided by the total number in the category was calculated as the RichFactor.

For the analysis of the human and rat lncRNA sequencing results, we used the UCSC liftOver tool to match the screened RNAs to the genome through collinear transformation. The conservation of RNA was also measured using the phylogenetic *p*-value conservation score. The conservation scores of each site of a lncRNA were counted separately, and the average of the conservation score values of all sites was taken as the conservation score of the region. To compare the conservation of a lncRNA and its upstream and downstream regions, we simultaneously screened the 2 kb upstream and downstream regions of the RNA for conservation scoring.

### 2.5. RNA–Protein Pulldown

We screened the conserved lncRNAs between humans and rats and constructed expression plasmids for these lncRNAs. Then, primers were designed to add a T7 promoter when amplifying linear DNA. DNA–RNA double-stranded molecules were obtained using a biotin-labeled UTP and MAXIscript T7 Transcription Kit (Thermo Fisher, Waltham, MA, USA) in vitro. Then, 2 µL of recombinant DNase I (RNase-free) was added, and the mixture was incubated at 37 °C for 15 min to obtain single-stranded biotin-labeled lncRNA. RNA was then purified using an RNeasy MinElute Cleanup Kit (QIAGEN, Germantown, MD, USA). Next, trypsin (0.25%) was added to digest astrocytes, which were subsequently lysed with RIPA and centrifuged at 12,000 rpm for 20 min to obtain a protein solution. The pretreated biotin RNA and 1 mg of the protein solution were mixed together, transferred to an enzyme-free EP tube, and incubated at 4 °C, protected from light, with rotation for 1 h. Then, avidin-conjugated magnetic beads were added, and the mixture was incubated overnight at 4 °C. Finally, the magnetic beads were washed with PBS, and protein was eluted with 2 × SDS buffer for identification via mass spectrometry.

### 2.6. Statistical Analysis

The results obtained in this study are expressed as the mean ± standard deviation. Significant differences between the two groups were compared using a *t*-test. *p* < 0.05 was considered statistically significant. GraphPad Prism 8.0 was used for experiment-related statistical analysis.

## 3. Results

### 3.1. OGD Induces Differential Expression of LncRNAs in Human BMEC-Derived EVs

We cultured human primary brain microvascular endothelial cells. Fifth-passage cells were subjected to OGD to simulate cerebral ischemic injury. The cell supernatant was collected to enrich EVs. After RNA was extracted, sequencing was performed. A total of 1335 lncRNAs were identified when the sequencing results were compared with the database. The chromosomal locations of these lncRNA genes were then analyzed. The Circos diagram in Figure 2A shows that these lncRNAs were distributed on all chromosomes. The fewest lncRNAs were distributed on the sex chromosomes (X and Y) and autosomes 4 and 13, while autosomes 16 and 17 harbored more lncRNA genes. The volcano plot shows the differentially expressed genes between the two groups of samples. Figure 2B shows the DE lncRNAs identified in the EVs of BMECs after OGD. The *x*-axis in the figure represents the fold-change values, and the *y*-axis represents the *p*-values. A total of 146 increased lncRNAs and 331 decreased lncRNAs were identified. Additionally, the expression level of each lncRNA is shown in Appendix A. Nine lncRNAs showed very low *p*-value and high fold change due to significant upregulation after OGD **(**Figure 2B). In Figure 2C, a heatmap is used to display the expression level of each lncRNA, with a redder color indicating increased lncRNA and a greener color indicating decreased lncRNAs. The upper panel shows the lncRNAs exhibiting elevated expression after OGD, and the lower panel shows the lncRNAs exhibiting reduced expression after OGD.

### 3.2. Functional Analysis of DE LncRNAs in Human BMEC-Derived EVs

After the identification of DE lncRNAs between the normal group and the OGD group, we predicted the target genes of these lncRNAs associated with the genomic upstream and downstream regions. We performed signaling pathway and GO enrichment analyses of these target genes. Figure 3 shows the top 10 identified signaling pathways and the GO enrichment analysis results. The sizes of the circles in the figure represent the numbers of lncRNA target genes enriched in the terms, with larger circles indicating more target genes of that term. The color of the circle represents the *p*-value of lncRNA target gene enrichment for that term. The number of genes in each category divided by the total number in that category was calculated as the RichFactor.

Figure 3A shows the enrichment results of the target genes of increased lncRNAs. The bubble plot on the left shows the results of the GO analysis, with the apoptotic process and intracellular signal transduction entries being the most significant. The bubble map on the right shows the results of pathway enrichment. Focal adhesion, the chemokine signaling pathway, the TNF signaling pathway, and the sphingolipid signaling pathway presented the highest enrichment scores among all terms, indicating that the increased lncRNAs may be functionally related to these terms. Among the reduced lncRNAs (Figure 3B), GO terms such as ATP binding, protein kinase binding, and protein serine/threonine kinase activity showed higher enrichment scores. These lncRNAs are mainly concentrated in the PI3K–Akt signaling pathway, MAPK signaling pathway, and cGMP–PKG signaling pathway.

### 3.3. OGD Induces Differential Expression of LncRNAs in Rat BMEC-Derived EVs

The sequence conservation of lncRNAs among species is relatively low. However, from an evolutionary perspective, lncRNAs that are important for carrying out a specific function in cells should be conserved among species. Therefore, we obtained primary rat brain microvascular endothelial cells via immune enrichment using the endothelial cell marker CD31. The OGD experiment was also performed using fifth-passage BMECs. The cell supernatant was collected, and EVs were extracted for lncRNA sequencing. After aligning the obtained sequence information with the database, we identified 400 lncRNAs, which were annotated in NCBI and Ensembl databases. In addition, we analyzed the coding potential of each transcript with Cuffquant software. Transcripts showing no coding potential according to the analysis results were predicted to be new lncRNAs. Through the analysis, we obtained a total of 4258 new lncRNAs. Next, we performed chromosomal mapping of all identified lncRNAs. The Circos diagram in Figure 4A shows that the lncRNAs identified with sequencing were distributed on all chromosomes. The fewest lncRNAs were distributed on sex chromosome Y, while the greatest number were distributed on autosome 1. We next analyzed the DE lncRNAs present in BMEC EVs after OGD and created a volcano plot (Figure 4B). Similar to the plot described above, the *x*-axis in the figure represents the fold-change values, and the *y*-axis represents the *p*-values. Red dots indicate upregulated lncRNAs (fold change > 1.5). Green dots indicate downregulated lncRNAs (fold change > 1.5). Gray dots indicate lncRNAs showing no difference in expression. For lncRNAs exhibiting a more than 1.5-fold difference in expression, we provide a separate heatmap in Figure 4C. The top panel shows new lncRNAs that were increased after OGD (a total of 1130); the next panel shows lncRNAs that were decreased (a total of 1114); the next panel shows annotated lncRNAs that were upregulated after OGD (a total of 85), and the bottom panel shows lncRNAs with downregulation (a total of 86). Detailed information on these lncRNAs is shown in Appendix A. Again, in the squares corresponding to each lncRNA in the figure; a redder color indicates higher expression levels, and a greener color indicates lower expression levels.

### 3.4. Functional Analysis of DE LncRNAs in Rat BMEC-Derived EVs

Next, we performed a functional analysis of DE lncRNAs between the normal and OGD groups. First, we performed chromosomal localization of the lncRNAs and then screened the possible coexpressed target genes of the lncRNAs within a range of 100 kb. Based on the bioinformatics analysis of these genes, signaling pathway and GO enrichment analyses were performed. Figure 5 shows the results of the top 10 signaling pathways and the GO enrichment analysis of the target genes of DE lncRNAs. The sizes of the circles in the figure represent the numbers of target genes for each term. The color of the circle represents the *p*-value of target gene enrichment for that term. The RichFactor represents the proportion of target genes among the total genes of each term, where the larger the value, the greater the number of target genes related to that term.

Figure 5A shows the enrichment results of increased lncRNA target genes. The bubble chart on the left shows the GO analysis results, with the apoptotic process and protein phosphorylation entries being the most significant. The bubble diagram on the right shows the results of pathway enrichment. Vascular smooth muscle contraction, the TNF signaling pathway, and the Wnt signaling pathway exhibited the highest enrichment scores among all entries. These findings indicated that the increased lncRNAs may be functionally related to these items. Among the decreased lncRNAs (Figure 5B), GO entries such as ATP binding, RNA binding, and ATPase activity exhibited higher enrichment scores. These lncRNAs were mainly involved in the regulation of the actin cytoskeleton, MAPK signaling pathway, and cAMP signaling pathway.

Further analysis revealed that the apoptotic process term was predicted based on both the human and rat results in the GO analysis. Vascular smooth muscle contraction, the TNF signaling pathway, the sphingolipid signaling pathway, and the leukocyte transendothelial migration pathway were also predicted in both the human and rat samples. These findings indicated that the upregulated lncRNAs may be involved in the apoptosis and inflammatory response of endothelial cells. Similarly, the GO results of the lncRNAs with downregulation identified ATP binding and ATPase activity among both the human and rat data, and the MAPK signaling pathway, regulation of the actin cytoskeleton, cAMP signaling pathway, and insulin signaling pathway all appeared in the top 10 results. These findings indicated that these decreased lncRNAs may be related to metabolism and cell growth.

### 3.5. Protein Binding Analysis of Conserved DE LncRNAs

Next, we analyzed the conservation of the DE lncRNAs in humans and rats. These highly conserved lncRNAs may have important functions after cerebral ischemia. The results showed that these DE lncRNAs were not highly conserved in humans and rats. Among these lncRNAs, 23 human BMEC-derived lncRNAs and 11 rat lncRNAs presented conserved sequences (from 1–92%). These lncRNAs constituted a total of 32 pairs of lncRNAs with conserved sequences (Table 1). Detailed information on these lncRNAs is shown in Appendix A. We further screened and eliminated sequence pairs with less than 50% conservation between humans and rats and obtained eight human lncRNAs and five rat lncRNAs. RNAs were then extracted from EVs, and reverse transcription PCR amplification was performed to confirm their expression changes after OGD. Finally, we screened six lncRNAs whose expression changes corresponded to the sequencing results that were conserved between humans and rats: NR_002323.2, NR_145459.1, and NR_144567.1 of human origin and ENSRNOT00000076420.2, ENSRNOT00000093096.1, and ENSRNOT00000092869.1 of rat origin (Figure 6A). These lncRNAs formed five pairs of conserved lncRNAs. We used the online BLAST tool to perform sequence alignment, revealing the conserved sequence information of these lncRNAs (Figure 6B). In this figure, red indicates conserved regions. We then added recombinant plasmids to upregulate human NR_002323.2, NR_145459.1, and NR_144567.1 in microglia cell line HMC3. TMEM119 antibody was used to confirm these cells were microglia via immunofluorescence. CD68 as the M1 activation marker was used to detect the microglia activation. The results showed the NR_002323.2 overexpression in HMC3 induced the M1 activation (Figure 6C). This suggests that the activation of microglia after cerebral ischemia may be related to EVs released by endothelial cells. Moreover, we used PCR to amplify the three conserved *Homo sapiens* lncRNAs. These lncRNAs were subsequently cloned into plasmids, and transcriptional promoters were subsequently added to the sequences. In vitro transcription and biotin labeling were performed to obtain a large number of biotin-conjugated lncRNA sequences. These lncRNAs were incubated with cell protein lysates of astrocytes and enriched using avidin-conjugated magnetic beads, and the binding proteins of the lncRNAs were identified via mass spectrometry. Table 2 shows the top three possible binding proteins for each lncRNA score. The binding of these proteins to lncRNAs may play a role in the regulation of mRNA translation. In conclusion, lncRNAs in vesicles released by endothelial cells may regulate mRNA translation in astrocytes.

## 4. Discussion

In this study, we subjected human and rat primary BMECs to OGD. Extracellular vesicles in the medium were enriched to detect the changes in lncRNAs. After bioinformatics analysis, we found that the increased lncRNAs identified in humans and rats may be related to apoptotic processes, vascular smooth muscle contraction, the TNF signaling pathway, the sphingolipid signaling pathway, and leukocyte transendothelial migration. When an ischemic stroke occurs, calcium ions in mitochondria are overloaded, causing damage to mitochondrial functions, the release of cytochrome C into cells, the activation of Caspase-3 and related cascade reactions, damage to DNA, and ultimately cell apoptosis, leading to a series of bodily dysfunctions [17]. Some studies have reported that in rats subjected to cerebral ischemia, inhibiting the expression of the proapoptotic factor Caspase-3 can reduce the volume of cerebral infarction and exert a protective effect on brain tissue [18]. In this study, human NR_002323.2 and rat ENSRNOT00000076420.2 were upregulated after OGD treatment (Figure 5A). They were annotated as taurine upregulated 1 (TUG1) in the database. Chen. et al. reported that knockdown of TUG1 decreased the ratio of apoptotic cells and promoted cell survival in vitro, which may be regulated by the elevated miRNA-9 expression and decreased Bcl2l11 protein [19]. After OGD, the expression of lncRNA NR_120420 was significantly increased in cells, which promoted cell apoptosis. After the knockout of NR_120420, the phosphorylation of NF-κB was significantly decreased, and the apoptosis rate was also significantly decreased [20]. Therefore, the inhibition of cell apoptosis will be the focus of future research aimed at controlling the development of brain damage after an ischemic stroke.

Furthermore, inflammatory responses are observed throughout the occurrence of ischemic strokes and are a key component of the pathogenesis of cerebral ischemic reinjury. In the early stage of ischemic strokes, microglia and astrocytes are activated, and cytokines and chemokines reach the penumbra through the activated blood vessel wall to produce inflammatory factors. In the later stage of ischemic strokes, there are many inflammatory factors in the penumbra. Studies have shown that blocking the expression of inflammation-related cytokines exerts a protective effect on cerebral-ischemia-induced brain injury in rats [21]. In addition, the functions of lncRNAs in vesicles released by endothelial cells are increasingly being revealed. Here, the lentivirus was used to upregulate the H_NR_002323.2 in human microglial HMC3 cells, which induced microglial M1 activation (Figure 5). Moreover, endothelial-progenitor-cell-derived EVs were shown to transmit lncRNA TUG1 to promote M2 macrophage polarization [22]. Additionally, TUG1 also promotes endothelial cell proliferation and increases angiogenesis [23]. Therefore, the lncRNAs released by endothelial cells identified in our study may regulate brain inflammation and apoptosis after ischemia by entering neurons and glial cells through EVs.

The decreased lncRNAs may be related to ATP binding, ATPase activity, the MAPK signaling pathway, the regulation of the actin cytoskeleton, the cAMP signaling pathway, and the insulin signaling pathway. This suggests that lncRNAs with downregulation may be associated with mitochondrial function. Mitochondria are one of the most important energy supply units in brain tissue. When brain tissue undergoes ischemia and hypoxia, the electron transport chain is destroyed and cannot release sufficient energy to drive the phosphorylation of ADP to generate ATP, resulting in an energy generation disorder [24]. Additionally, the content of lactate in the brain tissue of middle cerebral artery embolism (MCAO) model rats was increased, and the citric acid was decreased, indicating the defect in the tricarboxylic acid (TCA) after cerebral ischemia [25]. Here, mitochondrially localized lncRNA growth-arrest-specific 5 (GAS5) was downregulated in OGD cells (Appendix A). It negatively correlates with levels of its associated mitochondrial tricarboxylic acid (TCA) metabolic enzymes [26]. Additionally, NR_144567.1 was also downregulated in ODG cells (Figure 5A). The NR_144567.1 was annotated as metastasis-associated lung adenocarcinoma transcript 1 (MALAT1). The targeted knockdown of MALAT1 in endothelial cells inhibited the glucose uptake and exacerbated oxidative stress, attenuating angiogenesis [27,28]. The MALAT1 antisense RNA (TALAM1) expressed at the same position as the MALAT1 gene was identified as the conserved lncRNA NR_145459.1 in the present study. It was also downregulated in endothelial cells (Figure 5A). The depletion of TALAM1 leads to compromises in the cellular accumulation of MALAT1 [29]. Therefore, the release of lncRNAs by endothelial cells that we identified in our study was reduced after injury, reducing the regulation of mitochondrial energy.

RNA binding proteins (RBPs) combine with RNAs to form ribonucleoprotein (RNP) complexes, which play roles in all aspects of mRNA functions, including translation regulation, mRNA precursor splicing, nuclear export, and mRNA localization [30]. It has been reported that most lncRNAs can interact with corresponding RBPs and play irreplaceable roles in various cellular physiological processes [31]. In this study, RNA pulldown enriched RENT1 as the binding protein of NR_002323.2. RENT1 is also named UPF1, which is a key player in nonsense-mediated mRNA decay (NMD) but is also involved in posttranscriptional gene regulation. The inhibition of RENT1/UPF1 increased the steady-state expressions of ATF-4, ATF-3, and CHOP mRNAs, and many of the mRNAs for components of the integrated stress response have characteristics that could render them sensitive to NMD [32]. During apoptosis, UPF1 is cleaved by Caspases 3 and 7. The functional consequences of these cleavages are a general shutdown of NMD activity, leading to the stabilization of both premature termination codon-containing mRNAs and natural substrates of NMD, and also the production of caspase-cleaved UPF fragments that induce apoptosis and inhibit NMD [33]. Heterogeneous nuclear ribonucleoprotein U (hnRNPU), as the binding protein of NR_145459.1, has been reported to interact with lncRNA to regulate the transcription of p21 and control the stability of mRNA, thereby promoting lncRNA enrichment in the nucleus [34]. In addition, it has been confirmed that hnRNPU is abundant in EVs and localizes to the nucleus after entering the cell. The binding of hnRNPU to miR-30c-5p reduces HCAEC migration and promotes angiogenesis [35]. Moreover, NR_144567.1-bound protein RBFOX2 has a well-characterized role in the alternative splicing (AS) regulation of pre-mRNAs that can affect gene expression and function. RBFOX2 controls AS by binding to a highly conserved motif, (U)GCAUG, in intronic and/or exonic regions of pre-mRNAs [36]. The depletion of RBFOX2 adversely affects mitochondrial health by Slc25a4 inhibition. Slc25a4 gene encodes for ATP/ADP translocator 1 (ANT1), which is critical for energy production in the mitochondria [37]. It has been shown that ablation of Slc25a4 (ANT1) in mice exhibited a severe defect in coupled mitochondrial respiration [38].

In conclusion, the expression of lncRNAs in vesicles released by endothelial cells is altered after cerebral ischemic injury and may be involved in various cellular functions. Altered lncRNAs may be important in the pathogenesis of cerebral ischemic injury and may consequently represent potential therapeutic targets for treating cerebral ischemic diseases.

## Figures and Tables

**Figure 1 brainsci-12-01027-f001:**
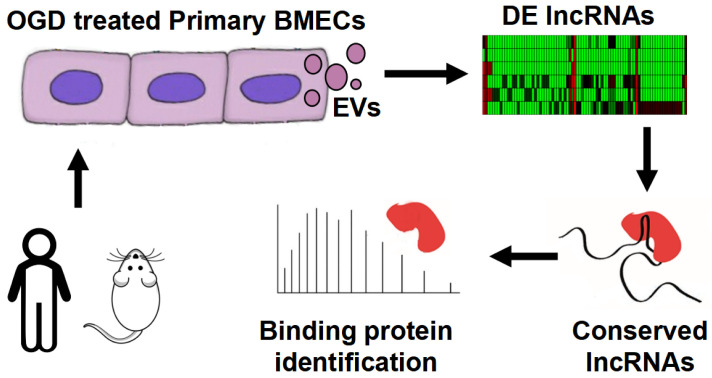
A schematic representation of study design. Oxygen and glucose deprivation (OGD)-induced lncRNA expression was identified via RNA sequencing in human and rat primary BMEC-released EVs. Additionally, potential functions and conservation analysis of the differentially expressed (DE) lncRNAs were subsequently performed. Finally, their binding proteins were identified via RNA pulldown and mass spectrometry using astrocytes.

**Figure 2 brainsci-12-01027-f002:**
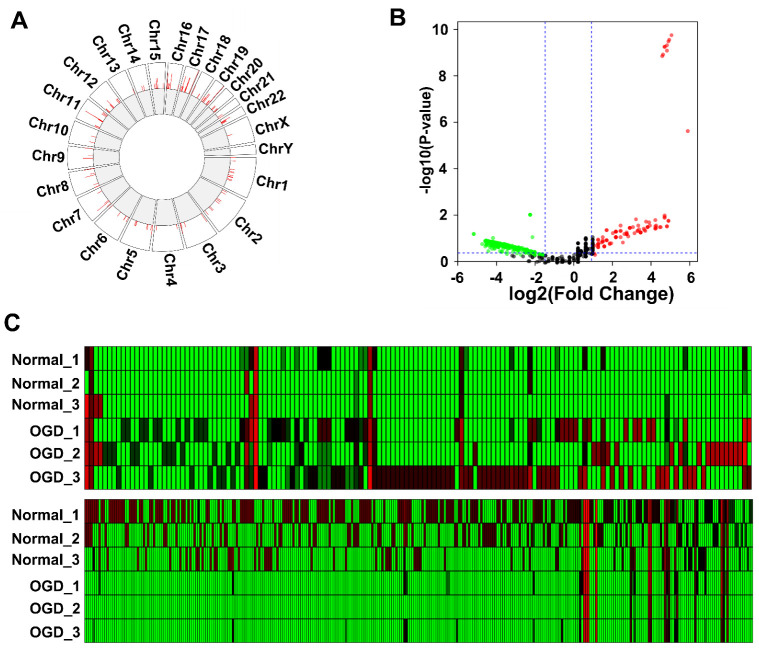
LncRNA sequencing of extracellular vesicles of human BMECs after oxygen and glucose deprivation (OGD): (**A**) Circos diagram showing the distribution of the lncRNAs identified by sequencing on chromosomes; (**B**) volcano plot of the identified lncRNAs. Red dots represent lncRNAs that were increased by 1.5-fold after OGD, green dots represent lncRNAs that were decreased by 1.5-fold, and gray dots represent lncRNAs with no significant difference in expression; (**C**) heatmap of significantly differentially expressed lncRNAs.

**Figure 3 brainsci-12-01027-f003:**
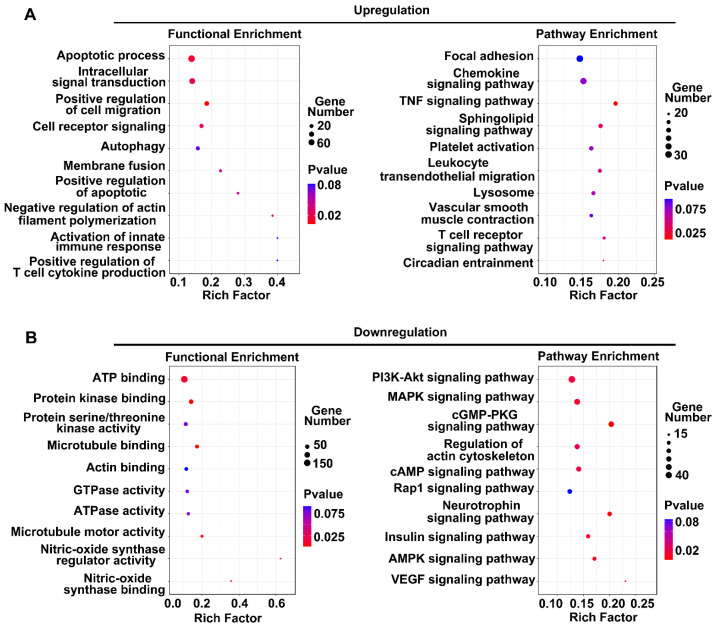
Functional prediction of significantly different lncRNAs in human BMECs: (**A**) functional enrichment analysis of coexpressed genes of upregulated lncRNAs on chromosomes. The left side shows the GO analysis results, and the right side shows the signal pathway analysis results; (**B**) analysis of the functions of coexpressed genes of downregulated lncRNAs.

**Figure 4 brainsci-12-01027-f004:**
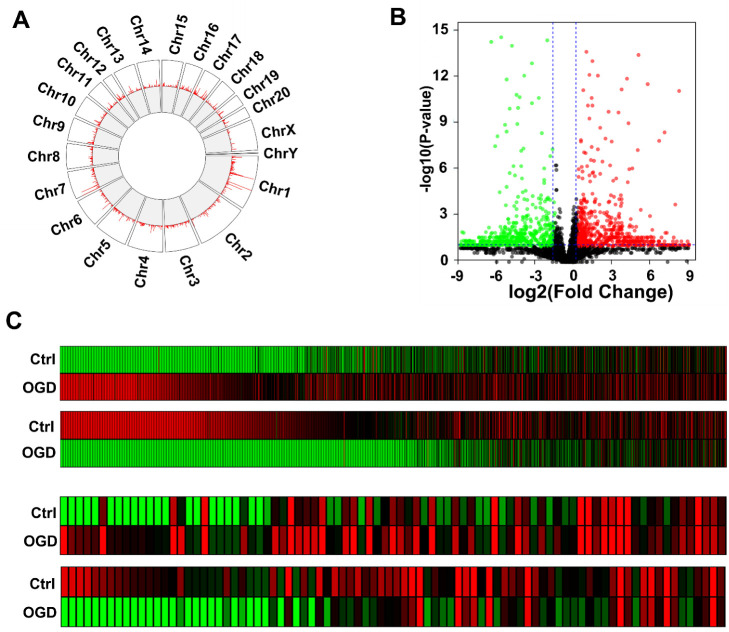
LncRNA sequencing of extracellular vesicles of rat BMECs after OGD: (**A**) Circos plot showing the distribution of lncRNAs identified via sequencing on chromosomes; (**B**) volcano plot of the identified lncRNAs. Red dots represent lncRNAs that were increased by 1.5-fold after OGD, green dots represent lncRNAs that were decreased by 1.5-fold, and gray dots represent lncRNAs with no significant difference in expression; (**C**) heatmap of significantly differentially expressed lncRNAs.

**Figure 5 brainsci-12-01027-f005:**
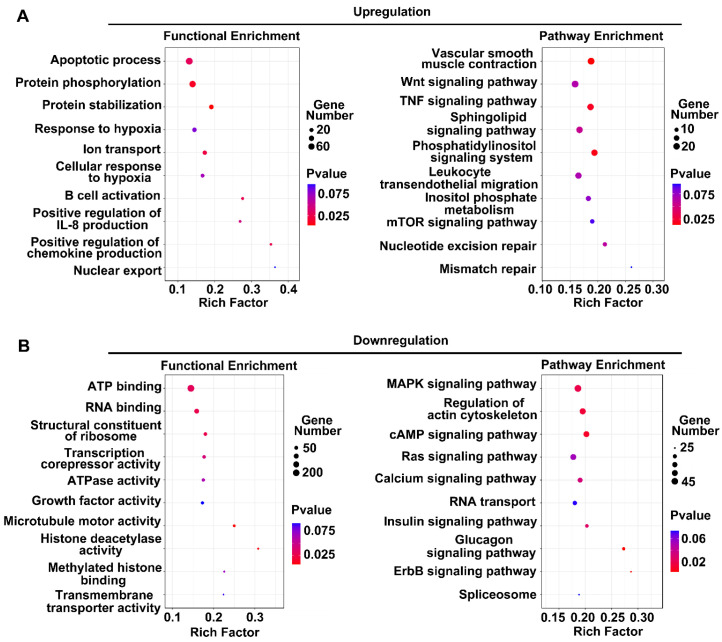
Functional prediction of significantly differentially expressed lncRNAs in rat BMECs: (**A**) functional enrichment analysis of upregulated lncRNAs. The left side shows the GO analysis results, and the right side shows the signal pathway analysis results; (**B**) analysis of the functions of coexpressed genes of downregulated lncRNAs.

**Figure 6 brainsci-12-01027-f006:**
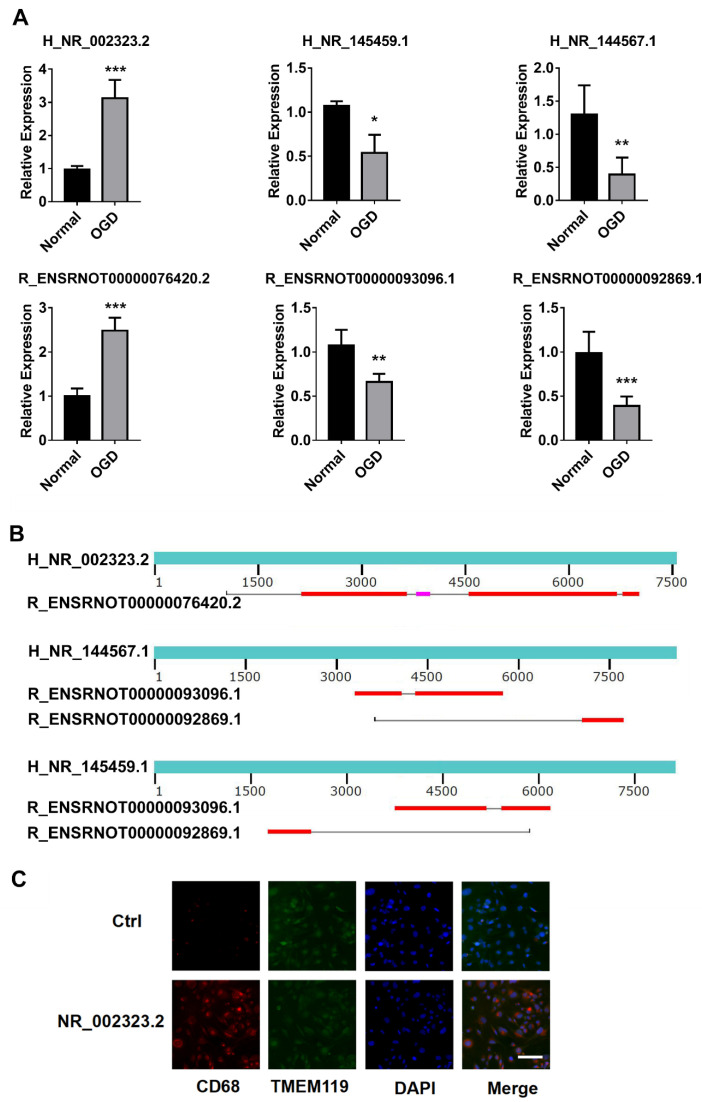
Analysis of conserved lncRNAs: (**A**) PCR confirmation of the expression of conserved human and rat lncRNAs; (**B**) sequence analysis of conserved RNAs; (**C**) immunofluorescence shows overexpression of NR_002323.2 in HMC3 cells (microglia) promotes M1 activation. TMEM119 is microglia marker. CD68 is M1 activation marker. *, compared with the normal group; * *p* ˂ 0.05, ** *p* ˂ 0.01, *** *p* ˂ 0.001.

**Table 1 brainsci-12-01027-t001:** Conservative analysis results of differential lncRNA between humans and rats.

H_LncRNA	Conservation%	R_LncRNA	Conservation%
NR_002323.2	62.42	ENSRNOT00000076420.2	91.94
NR_110493.1	62.35	ENSRNOT00000076420.2	91.66
NR_110492.1	61.07	ENSRNOT00000076420.2	90.61
NR_145459.1	26.83	ENSRNOT00000093096.1	92.72
NR_144567.1	25.5	ENSRNOT00000093096.1	92.72
NR_002819.4	24.82	ENSRNOT00000093096.1	92.72
NR_144568.1	23.73	ENSRNOT00000093096.1	83.83
ENST00000537157.1	20.88	ENSRNOT00000090171.1	10.28
NR_002809.2	13.15	ENSRNOT00000090171.1	20.55
NR_026794.1	8.32	ENSRNOT00000087244.1	45.89
NR_026794.1	8.32	ENSRNOT00000081933.1	38.66
NR_144568.1	7.72	ENSRNOT00000092949.1	92.23
ENST00000567556.1	7.66	ENSRNOT00000035590.6	1.72
NR_144567.1	7.5	ENSRNOT00000092949.1	92.23
NR_145459.1	7.34	ENSRNOT00000092869.1	84.06
NR_002819.4	7.3	ENSRNOT00000092949.1	92.23
NR_144568.1	7.18	ENSRNOT00000092869.1	84.06
NR_144567.1	6.97	ENSRNOT00000092869.1	84.06
NR_002819.4	6.79	ENSRNOT00000092869.1	84.06
NR_109854.1	6.08	ENSRNOT00000090651.1	11.1
NR_109853.1	5.96	ENSRNOT00000090651.1	11.1
NR_145459.1	5.1	ENSRNOT00000092949.1	59.57
NR_015431.2	4.38	ENSRNOT00000090651.1	11.1
NR_109852.1	4.24	ENSRNOT00000090651.1	11.1
NR_109851.1	4.21	ENSRNOT00000090651.1	11.1
NR_109850.1	4.21	ENSRNOT00000090651.1	11.1
NR_109855.1	3.45	ENSRNOT00000090651.1	4.29
NR_102327.1	1.43	ENSRNOT00000086884.1	3.16
NR_024053.2	0.7	ENSRNOT00000086884.1	3.16
NR_024052.2	0.67	ENSRNOT00000086884.1	3.16
NR_102326.1	0.54	ENSRNOT00000086884.1	3.16
NR_125909.1	0.14	ENSRNOT00000081606.1	1.63

**Table 2 brainsci-12-01027-t002:** Proteome identification results after lncRNA pulldown.

Probe	Accession	Name	MW [kDa]	Score	Coverage%	Peptides
NR_002323.2	Q92900	RENT1	124.3	100.23	3.84	3
	Q08170	SRSF4	56.7	160.7	5.18	2
	Q6NTF9	RHBD2	39.2	378.08	13.64	5
NR_145459.1	Q00839	HNRPU	88.9	128.85	2.95	2
	Q9NR90	DAZ3	54.9	175.25	10.18	3
	P55795	HNRH2	49.2	107.67	5.74	2
NR_144567.1	O43251	RFOX2	41.3	291.68	37.11	8
	A6NDY0	EPAB2	30.3	106.23	6.04	2
	Q9UJJ7	RUSD1	34.7	217.54	25.65	4

## Data Availability

The datasets used or analyzed during the current study are available from the corresponding author.

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
