# Peer review of "LncRNA-Profile-Based Screening of Extracellular Vesicles Released from Brain Endothelial Cells after Oxygen–Glucose Deprivation"

_brainsci, 2022, doi:10.3390/brainsci12081027_

Round 1
Reviewer 1 Report
The paper by He et al studies the interesting lncRNAs found in EVs secreted by endothelial cells during ischemia. LncRNAs are not that studied yet but poses interesting functions in cell and protein regulation. Unfortunately, the biological impact of the identified lncRNAs have not been studied in the present paper.
Major:
The figures are missing figure legends and captions. Several parts of the Results description should be moved to the figure legend thereby making the Results section easier to read.
The use of high/low expression is quite confusing as this does not refer to DE lncRNAs. This should be described as increased/reduced during ischemia.
Even though the bioinformatic analysis including the enrichment analysis is interesting. All this data is a bit speculative as no direct biological function in regards to stroke and/or ischemia is presented. The possible impact of 6 lncRNAs on astrocyte protein expression is only tested in relation to the binding of astrocytic proteins and not in vitro/in vivo astrocytes. It would be interesting to see if the identified lncRNAs would have a direct impact on living astrocytes (or other cells in the brain) by administering the lncRNAs to cultured cells and then quantify targeted protein levels.
In the discussion it is not clear whether the identified DE lncRNAs are protective of the brain tissue or causes and maybe even aggravates damage induced by ischemia. For example in the mitochondrial function discussion, how would the changes in lncRNAs impact energy turnover and how will that then impact cell viability during ischemia? Is it good for the cells to receive these lncRNAs from endothelial cells or do they
Line 161. Mice?
Line 184. Please state the temperature of ultra centrifugation
Line 270. Results section. Please include figure legends (eg. Fig 1). It would be better to summarize the findings in the text than explaining the volcano plot (this should be in the legend). It is much more interesting to read about how many lncRNAs are down- or upregulated as well as the very high fold change of some of the upregulated lncRNAs (is this due to very low expression in normoxic cultures?). In addition, information about the copy number of the DE lncRNA (are they abundantly expressed or more rare?)
Please use “downregulation / upregulation” when describing DE and not highly expressed / low expression, as the latter terms does not elute to the expression changes between the 2 conditions. This is also an issue for figure 2
Line 325. Why did you only perform coding potential of the rat lncRNAs?
Line 332: Please move the details of the volcano plot to a figure legend
Line 411. Several statements are missing references. Eg. molecular reactions to stroke, mitochondrial function, MAPK pathway. It is also unclear how some of the described stroke pathophysiological pathways relate to the identified lncRNAs.
Line 475. This is the most interesting part of the discussion together with the discussion of the protein pulldown as it directly discusses the possible function/impact of the 3 of the conserved lncRNAs. However, it could need a rewriting as it is not clearly formulated which lncRNA is involved and how they control protein expression.
Line 500: Why are you discussing Rbfox-1? It is only Rfox2 that is pulled down by lncRNA NR_144567.1.
Author Response
Thanks for the valuable suggestion. Our replies are as follows.
Comment 1:
The figures are missing figure legends and captions. Several parts of the Results description should be moved to the figure legend thereby making the Results section easier to read.
Reply: We are so sorry that missed the figure legends. We have added it at the end of the manuscript. Maybe editor will place it below the figures.
Comment 2:
The use of high/low expression is quite confusing as this does not refer to DE lncRNAs. This should be described as increased/reduced during ischemia.
Reply: Thanks so much for suggestion. We have corrected the expression.
Comment 3:
Even though the bioinformatic analysis including the enrichment analysis is interesting. All this data is a bit speculative as no direct biological function in regards to stroke and/or ischemia is presented. The possible impact of 6 lncRNAs on astrocyte protein expression is only tested in relation to the binding of astrocytic proteins and not in vitro/in vivo astrocytes. It would be interesting to see if the identified lncRNAs would have a direct impact on living astrocytes (or other cells in the brain) by administering the lncRNAs to cultured cells and then quantify targeted protein levels.
Reply: We up-regulated human NR_002323.2, NR_145459.1, and NR_144567.1 in microglia cell line HMC3. TMEM119 antibody was used to confirm these cells were microglia by Immunofluorescence. CD68 as the M1 activation marker was used to detect the microglia activation. The results showed the NR_002323.2 overexpression in HMC3 induced the M1 activation (Figure 6C).
Comment 4:
In the discussion it is not clear whether the identified DE lncRNAs are protective of the brain tissue or causes and maybe even aggravates damage induced by ischemia. For example in the mitochondrial function discussion, how would the changes in lncRNAs impact energy turnover and how will that then impact cell viability during ischemia? Is it good for the cells to receive these lncRNAs from endothelial cells or do they
Reply: We have re-written the discussion and added this information in the first to third paragraphs.
Comment 5:
Line 161. Mice?
Reply: This is a spelling mistake. We have corrected it to rats.
Comment 6:
Line 184. Please state the temperature of ultra centrifugation
Reply: We have added it as 4°C.
Comment 7:
Line 270. Results section. Please include figure legends (eg. Fig 1). It would be better to summarize the findings in the text than explaining the volcano plot (this should be in the legend). It is much more interesting to read about how many lncRNAs are down- or upregulated as well as the very high fold change of some of the upregulated lncRNAs (is this due to very low expression in normoxic cultures?). In addition, information about the copy number of the DE lncRNA (are they abundantly expressed or more rare?)
Reply: We have added figure legends in the manuscript.
After OGD, 146 increased lncRNAs and 331 decreased lncRNAs were identified. There are 9 lncRNAs showed very low p-value and high fold change due to significant up-regulation after OGD. The expression level of each lncRNA is shown in Supplementary Table 1. We have added this information in the first paragraph of the result.
Comment 8:
Please use “downregulation / upregulation” when describing DE and not highly expressed / low expression, as the latter terms does not elute to the expression changes between the 2 conditions. This is also an issue for figure 2
Reply: We have corrected the expression.
Comment 9:
Line 325. Why did you only perform coding potential of the rat lncRNAs?
Reply: The coding potential analysis was used to identify the novel lncRNAs that were not recorded in the database. When we performed the RNA sequencing of the human cells, we haven’t intended to do a conservation analysis with the rats. Second is the human lncRNAs database has collected the amount of lncRNAs information than rats. When we analyzed the data of human lncRNAs sequencing data, we recognized that the lncRNAs are poor conservation with different races. So we had to do the coding potential of the rat lncRNA so that we could obtain more lncRNAs of rat to do conservation analysis with human.
Comment 10:
Line 332: Please move the details of the volcano plot to a figure legend
Reply: We have added figure legends in the manuscript.
Comment 11:
Line 411. Several statements are missing references. Eg. molecular reactions to stroke, mitochondrial function, MAPK pathway. It is also unclear how some of the described stroke pathophysiological pathways relate to the identified lncRNAs.
Reply: We have removed these from the discussion.
Comment 12:
Line 475. This is the most interesting part of the discussion together with the discussion of the protein pulldown as it directly discusses the possible function/impact of the 3 of the conserved lncRNAs. However, it could need a rewriting as it is not clearly formulated which lncRNA is involved and how they control protein expression.
Reply: We have rewritten this section.
Comment 13:
Line 500: Why are you discussing Rbfox-1? It is only Rfox2 that is pulled down by lncRNA NR_144567.1.
Reply: There is little research about Rfox2. We have removed this section.
Reviewer 2 Report
1. Initial part of the experiments and methodology are difficult to follow for the readers. Please provide a schematic representation of study design as figure 1
2. The following sentence in the introduction should be removed. “Therefore, over billions of years of evolution, a blood–brain barrier that separates blood from brain tissue evolved”
3. Please provide animal ethics approval details for the study
4. How do you consider 20% FBS is required for this culture. Don’t you think it overshadows glucose deprivation? Apart from your previous paper, are there any references to this model of OGD?
Author Response
Thanks very much for your constructive comments. Our replies are as follows.
Comment 1:
Initial part of the experiments and methodology are difficult to follow for the readers. Please provide a schematic representation of study design as figure 1
Reply: We have added it at the end of the manuscript. Maybe editor will place it in the appropriate place of the manuscript.
Comment 2:
The following sentence in the introduction should be removed. “Therefore, over billions of years of evolution, a blood–brain barrier that separates blood from brain tissue evolved”
Reply: We have deleted this sentence.
Comment 3:
Please provide animal ethics approval details for the study
Reply: We have added it in the methods section.
Comment 4:
How do you consider 20% FBS is required for this culture. Don’t you think it overshadows glucose deprivation? Apart from your previous paper, are there any references to this model of OGD?
Reply: We have tested the glucose concentration after adding the 20% FBS in the glucose-free medium. It was too low to be detected. The paper that was published in 2019 (doi: 10.4103/1673-5374.262589) that was used the same model.
Reviewer 3 Report
Journal: Brain Sciences
Manuscript ID: brainsci-1777172
Type of manuscript: Article
Title: LncRNA profile-based screening of extracellular vesicles released from brain endothelial cells after oxygen-glucose deprivation.
Dear Editors,
In the present study, the Authors subjected primary human and rat BMECs to oxygen and glucose deprivation (OGD) and enriched extracellular vesicles (EV)s for RNA sequencing. They compared the sequencing results with the annotated database of known lncRNAs and revealed 146 upregulated lncRNAs and 331 downregulated lncRNAs in human cells and 85 upregulated lncRNAs and 86 downregulated lncRNAs in rat cells. Next, they analyzed the genes that were coexpressed with the differentially expressed (DE) lncRNAs on chromosomes and performed Gene Ontology (GO) and signaling pathway enrichment analyses. Their results showed that the lncRNAs may play roles in apoptosis, the TNF signaling pathway and leukocyte transendothelial migration. Then, they analyzed and confirmed three conserved lncRNAs between humans and rats by PCR. Moreover, they identified the binding proteins of these three lncRNAs in human astrocytes via RNA pull-down and mass spectrometry and found that these proteins could regulate mRNA stability and translation. In conclusion, they suggested that the identified lncRNAs with altered expression in EVs may be involved in intercellular signal transduction from BMECs to other cells after cerebral ischemia.
The findings mentioned above are interesting, and the manuscript has been written logically, with a highly satisfactory level of profession. In the following, certain minor comments are directed. In conclusion, the mentioned paper deserves to be published in this Journal provided that a minor revision is undertaken.
Kind regards,
1. A clear statement of the aim of the study is lacking throughout the whole text. Therefore, the Authors should explain their rationale or aim of the study both in the “Introduction” and “Abstract” sections.
2. The “Introduction” section has been written in a manner similar to a review article, so contains excessively detailed information. Therefore, I recommend the Authors consider shortening the text avoiding the unnecessary details in the “Introduction” section as well as in their “Abstract”.
3. All abbreviations should be checked throughout the text and they should be written in full upon the first appearance, for instance as long noncoding RNAs (lncRNAs).
Author Response
Thanks very much for your kind letter and comments. Our replies are as follows.
Comment 1:
A clear statement of the aim of the study is lacking throughout the whole text. Therefore, the Authors should explain their rationale or aim of the study both in the “Introduction” and “Abstract” sections.
Reply: We have re-written the “Introduction” and “Abstract” sections.
Comment 2:
The “Introduction” section has been written in a manner similar to a review article, so contains excessively detailed information. Therefore, I recommend the Authors consider shortening the text avoiding the unnecessary details in the “Introduction” section as well as in their “Abstract”.
Reply: We have re-written them to make them more clearly.
Comment 3:
All abbreviations should be checked throughout the text and they should be written in full upon the first appearance, for instance as long noncoding RNAs (lncRNAs).
Reply: We have checked and corrected these.